# Novel QTL Associated with Shoot Branching Identified in Doubled Haploid Rice (*Oryza sativa* L.) under Low Nitrogen Cultivation

**DOI:** 10.3390/genes12050745

**Published:** 2021-05-14

**Authors:** Young-Ho Kwon, Nkulu-Rolly Kabange, Ji-Yun Lee, So-Myeong Lee, Jin-Kyung Cha, Dong-Jin Shin, Jun-Hyeon Cho, Ju-Won Kang, Jong-Min Ko, Jong-Hee Lee

**Affiliations:** Department of Southern Area Crop Science, National Institute of Crop Science, RDA, Miryang 50424, Korea; kwon6344@korea.kr (Y.-H.K.); rollykabange@korea.kr (N.-R.K.); minitia@korea.kr (J.-Y.L.); olivetti90@korea.kr (S.-M.L.); jknzz5@korea.kr (J.-K.C.); jacob1223@korea.kr (D.-J.S.); hy4779@korea.kr (J.-H.C.); kangjw81@korea.kr (J.-W.K.); kojmin@korea.kr (J.-M.K.)

**Keywords:** shoot branching, quantitative trait locus, nitrogen, KASP markers, Fluidigm markers, rice

## Abstract

Shoot branching is considered as an important trait for the architecture of plants and contributes to their growth and productivity. In cereal crops, such as rice, shoot branching is controlled by many factors, including phytohormones signaling networks, operating either in synergy or antagonizing each other. In rice, shoot branching indicates the ability to produce more tillers that are essential for achieving high productivity and yield potential. In the present study, we evaluated the growth and development, and yield components of a doubled haploid population derived from a cross between 93-11 (P1, *indica*) and Milyang352 (P2, *japonica*), grown under normal nitrogen and low nitrogen cultivation open field conditions. The results of the phenotypic evaluation indicated that parental lines 93-11 (P1, a high tillering *indica* cultivar) and Milyang352 (P2, a low tillering *japonica* cultivar) showed distinctive phenotypic responses, also reflected in their derived population. In addition, the linkage mapping and quantitative trait locus (QTL) analysis detected three QTLs associated with tiller number on chromosome 2 (*qTNN2-1*, 130 cM, logarithm of the odds (LOD) 4.14, PVE 14.5%; and *qTNL2-1*, 134 cM, LOD: 6.05, PVE: 20.5%) and chromosome 4 (*qTN4-1*, 134 cM, LOD 3.92, PVE 14.5%), with *qTNL2-1* having the highest phenotypic variation explained, and the only QTL associated with tiller number under low nitrogen cultivation conditions, using Kompetitive Allele-Specific PCR (KASP) and Fluidigm markers. The additive effect (1.81) of *qTNL2-1* indicates that the allele from 93-11 (P1) contributed to the observed phenotypic variation for tiller number under low nitrogen cultivation. The breakthrough is that the majority of the candidate genes harbored by the QTLs *qTNL2-1* and *qTNN4-1* (here associated with the control of shoot branching under low and normal nitrogen cultivation, respectively), were also proposed to be involved in plant stress signaling or response mechanisms, with regard to their annotations and previous reports. Therefore, put together, these results would suggest that a possible crosstalk exists between the control of plant growth and development and the stress response in rice.

## 1. Introduction

The global population is increasing at a relatively high growth rate per annum, and is projected to reach about 9.8 billion people by 2050 [1]. The current status of the food and nutrition security in the world question on how to provide sufficient, healthy and nutritious food, at all times to the growing population. To achieve this, several initiatives have been taken to combat malnutrition and food insecurity, and alleviate poverty and eradicate hunger, in line with the United Nations Sustainable Development Goals (SDGs), through research and development, among others.

Since the domestication of rice, this Poaceae is the second most important cereal crop after corn and remains today the only cereal crop solely cultivated for human consumption, in addition to being the staple food crop for more than half of the world’s population, despite the increasing preference for various foods and the diversification of diet. According to the Food and Agriculture Organization of the United Nations (FAO) statistics (http://www.fao.org/faostat/en/#data/QC/visualize, accessed on 5 March 2021), nearly 755.5 million metric tons of paddy rice were harvested in 2019, and around 162.1 million hectares were occupied by rice (upland, wetland, and irrigated) fields across the globe.

To achieve optimum plant growth and development, and realize their potential in terms of productivity and quality, plant crops require nitrogen (N), among other essential macronutrients [2]. The role of nitrogen in the metabolism of plants is important in promoting seed germination, cell division, plant growth and development, formation of amino acids and proteins, and plant productivity [3,4,5,6]. Rice cultivation requires abundant use of nitrogenous fertilizers [7], particularly in high yielding varieties [8]. However, excessive application of nitrogenous fertilizers has been associated with the emission of greenhouse gases (GHG), such as methane (CH_4_) and nitrous oxide (N_2_O) [9,10,11,12].

In the recent years, the trend in plant breeding revealed a growing interest in developing rice varieties with an optimized and balanced productivity and quality, while exhibiting a high nitrogen use efficiency (NUE), which may help reduce the use of large amounts of fertilizers, particularly the N-rich fertilizers or a combined mineral–organic fertilization regime [13,14,15,16], as part the efforts to reducing GHG emission from agricultural activities, and to promoting the use of ecofriendly agricultural practices. To achieve this, scientists have employed various strategies that include the use of advanced plant breeding techniques and the available and emerging technologies to develop rice varieties with an improved nitrogen use efficiency (NUE), which includes nitrogen uptake, transport, assimilation, and remobilization. It has been shown that an increase in nitrogen (N) fertilizers is associated with a large decrease in NUE [17]. Therefore, there is a pressing need to improve agricultural practices in order to contribute to the reduction of GHG emissions from irrigated, rainfed, and flood-prone rice ecosystems, to predict the future emissions and develop rice technologies that will reduce GHG emissions, and yet will meet the required increase in rice production [18]. 

In rice, shoot branching is considered as one of the major agronomic traits with a high potential to contribute to the yield and productivity of rice [19,20]. It is well established that phytohormones play fundamental roles in the control of shoot branching and bud outgrowth in plants, operating in a synergetic or antagonistic manner [21,22,23,24,25], in addition to their roles in seed dormancy and germination, plant growth and development, flowering and organogenesis, seed formation and maturation, fruit ripening, and signaling during abiotic or biotic stress occurrence [26]. Several studies have proposed a number of quantitative trait loci (QTLs) controlling tiller number in rice, and mapped to almost all rice chromosomes (10–11) using different sorts of mapping populations, including doubled haploid (DH) lines [27]. Many genes have been identified; however, only a few genes have been cloned and functionally characterized [28,29,30,31,32,33,34,35,36,37]. A mapping population derived from a cross between *indica* and *japonica* would offer the best cross combination to investigate QTLs associated with rice plant architecture, due to the morphological difference between *indica* and *japonica* subspecies.

The use of fixed homozygous populations, such as doubled haploid (DH) [38,39] and recombinant inbred lines (RILs) [40] have been favored when investigating QTLs for various agronomic traits in rice. The *indica* and *japonica* rice subspecies, are the most cultivated rice, and are said to be independently domesticated from the wild ancestor [41,42,43]. 

Therefore, from this perspective, this study performed a linkage mapping analysis on a doubled haploid (DH) developed through anther culture of a cross between 93-11 (P1) and Milyang352 (P2), typical *indica* and *japonica* rice cultivars, respectively, grown under normal nitrogen application and low nitrogen-level field conditions, to investigate putative QTLs controlling rice tiller number as well as agronomic traits contributing to the yield of rice using Kompetitive Allele-Specific PCR (KASP) and Fluidigm markers. Additionally, this study identified candidate genes associated with tiller number in rice under low nitrogen cultivation, with interesting annotated molecular functions, and suggested to have the potential to regulate shoot branching and panicle number in rice. 

## 2. Materials and Methods

### 2.1. Mapping Population, Growth Conditions, and Phenotypic Measurements

A total of 117 doubled haploid (DH) rice lines developed through another culture of the cross between *Oryza sativa* L. ssp. *indica* cv. 93-11 (P1) and *Oryza sativa* L. ssp. *japonica* cv. Milyang352 (P2), respectively, were used as the mapping population. Prior to sowing, seeds were soaked in 0.7% nitric acid (HNO_3_) CAS: 7697-37-2, Lot No. 2016B3902; Junsei Chemical Co. Ltd., Tokyo, Japan) for about 24 h to break the dormancy, followed by incubation at 32 °C for 48 h to induce germination. Then, germinated seeds were sown in 50-well trays containing an enriched soil (Doobaena Plus, Nong Kyung Ltd., Yeongcheon-si, Korea) until transplanting time. Seedlings with uniform height were transplanted in the experimental field located at the National Institute of Crop Science (NICS) (altitude: 11 m, 35°29′31.4″ N and 128°44′30.0″ E), Rural Development Administration (RDA), Miryang, Republic of Korea. The experiments were initially conducted during the rice cropping season (June–November) of 2019 and repeated in 2020 to validate the phenotypes prior to performing QTL analysis. One set of the mapping population and parental lines were transplanted on a plot with a normal nitrogen cultivation regime (90 kg ha^−1^ N, 45 kg ha^−1^ P_2_O_5_, and 57 kg ha^−1^ K_2_O), and the other on a plot with a low nitrogen level (45 kg ha^−1^ N, 22.5 kg ha^−1^ P_2_O_5_, and 28.5 kg ha^−1^ K_2_O) maintained over 15 years. Each rice line had at total of 100 seedlings transplanted in four rows, with 25 plants per row and the spacing between and within the lines of 30 cm × 15 cm, respectively. Parental lines were planted after every 20 DH lines, starting from the initial rows. The weather parameters and conditions during the experimental period can be found in Appendix A.

The soil chemical analysis was performed following the method described by Park et al. [44] and the basic soil chemical properties are given in Table 1.

The phenotypic measurements of the studied traits, which included plant height (PH), tiller number (TN), culm length (CL), panicle length (PL), number of panicles per plant (PN), root length (RL), shoot dry weight (SDW), and root dry weight (RDW), were done on ten individual plants in triplicate randomly pooled from the inside rows, excluding the border rows to avoid the border effects on the traits studied and competition between lines. The plant height and culm length were measured using a 1-m long wooden stick ruler at maximum tillering stage. The panicle length was measured with a regular ruler soon after harvesting. The number of tillers per plant was counted at maximum tillering stage. 

SDW was measured from an oven-dried (at about 45 °C overnight) separated shoot from the roots at the collar, cooled down, and weighed on a precision scale (CASKOREA Co. Ltd, MW2-6004, 0.02–60 g, Seongnam-si, Korea). Similarly, RDW was measured after washing off any loose soil particles, followed by blotting of samples with a soft paper towel to remove any free surface moisture. Samples were oven-dried, weighed, and the results expressed in grams.

### 2.2. Frequency Distribution, Quantile–Quantile Plots, Principal Component Analysis, and Correlation Analysis

The frequency distributions of each phenotypic trait under normal and low nitrogen cultivation and the correlation analysis were performed using GraphPad Prism software (version 7.00, © 1992–2016 GraphPad, San Diego, CA, USA). The normality of the distribution was assessed using the Shapiro–Wilk W-statistic for the test of normality in IciMapping (version 4.1.0.0, 2016, Chinese Academy of Agriculture Sciences, Beijing, China). In addition, the Quantile–Quantile (Q–Q) plots and the pairwise kinship matrix were generated using the GAPIT function: *my_GAPIT <-GAPIT(Y = myY*, *G = myG*, *Model.selection = TRUE*, *SNP.MAF = 0.05)* in RStudio (version 1.2.5042, © 2009–2020 RStudio, Inc., Boston, MA, USA). The cluster analysis was performed using the R packages *ggplot2*, *tidyverse*, *cluster*, and *fviz_cluster* R function. The principal component analysis (PCA) was performed using the R package *ggplot2* and functions *myPr*, *plot*, and *biplot* [45,46].

### 2.3. Genotyping and Molecular Markers Analysis

For the molecular markers analysis, Kompetitive Allele-Specific PCR (KASP) markers and Fluidigm markers were employed. KASP markers were amplified from genomic DNA and the allelic discrimination was performed using the Nexar system (LGC, Biosearch Technologies, Hoddesdon, UK), while the Fluidigm markers for SNP genotypes were determined using the BioMark^TM^ DH system (Fluidigm, San Francisco, CA, USA) and 96.96 Dynamic Array IFC (96.96 IFC) chip as described earlier [47].

### 2.4. Construction of Linkage Maps and QTL Analysis

A total of 240 markers, including KASP markers earlier reported [48] to be specific for detecting polymorphism between *japonica* subspecies and Fluidigm markers [49] specific for detecting polymorphism between *indica* and *japonica* subspecies, and the phenotypic data of the mapping population consisting of 117 doubled haploid (DH) rice lines and their parental lines (93-11, P1 and Milyang352, P2) were used to perform QTL analysis in order to identify putative QTLs associated with plant growth, tiller number, panicle length, panicle number, root length, and shoot and root dry weights in rice, under low nitrogen cultivation. The genotype and phenotype raw data were initially formatted in RStudio using the *fread*, *pheno.raw*, *geno.raw*, and *cbind* functions. The QTL analysis was performed and the linkage maps constructed with IciMapping software v.4.1.0.0, for a biparental population using position mapping and Kosambi mapping functions [50]. The permutation (1000 times) parameters, which explains the probability for detecting statistically significant QTLs associated with the target traits, were selected.

The rice genome annotation database publicly available online browser (http://rice.plantbiology.msu.edu/, accessed on 22 February 2021) was used to identify candidate genes pooled from the detected QTL *qTNL2-1* (684.4 kbp, Chr2:26706424..27390847) region flanked by KASP markers KJ02_057 (left, closest marker) and id2012773 (right marker) on chromosome 2, associated to the control of the number of tillers under low nitrogen cultivation. 

## 3. Results

### 3.1. Differential Growth Patterns between Doubled Haploid Lines under Normal and Low Nitrogen Conditions

We evaluated the phenotypic response of the mapping population and their parental lines under normal and low nitrogen cultivation conditions. The results revealed a normal distribution for plant height (Figure 1A,B, Appendix A, under normal and low nitrogen cultivation), number of panicles (Figure 1I, Appendix A, under normal nitrogen cultivation), and root length (Figure 1K,L, Appendix A, under normal and low nitrogen cultivation). However, for culm length (Figure 1F, under low nitrogen cultivation) a negative skewness was observed. In contrast, a positive skewness was recorded for tiller number (Figure 1C,D, Appendix A), culm length (Figure 1E, Appendix A, under normal nitrogen cultivation), panicle length (Figure 1G,H, Appendix A, under normal and low nitrogen cultivation), panicle number (Figure 1J, Appendix A, under low nitrogen cultivation), shoot dry weight (Figure 1M,N, Appendix A, under both normal and low nitrogen cultivation), and root dry weight (Figure 1O,P, Appendix A, under normal and low nitrogen cultivation). Similarly, the Quantile–Quantile (Q–Q) plots (observed quantiles plotted against theoretical quantile or z-scores) reflected the same distribution patterns of the traits observed earlier (Appendix A).

Our data indicate that about 8.5% and 91.5% of the DH lines had relatively short plants (P1-like phenotype) and tall plants (P2-like phenotype), respectively, under normal nitrogen conditions. Whereas, under low nitrogen cultivation, nearly 39.3% and 60.7% had P1-like (relatively short plants) and P2-like (relatively tall plants) plant height, respectively (Figure 2A,I,J). In addition, about 68.4% of the 117 DH lines grown under normal nitrogen conditions recorded an increase in number of tillers per plant similar to P1 (>13 tillers per plant) against 31.6% with P2-like tiller number (≤13 tillers per plant), while about 69.2% and 30.8% of the DH lines cultivated under low nitrogen level had P1-like and P2-like tiller number, respectively (Figure 2B). In the same way, about 83.8% of the mapping population exhibited an increase in culm length (P1-like, >58 cm) against 14.5% that showed a P2-like culm length (≤58 cm) under normal nitrogen cultivation. However, when the DH lines were grown under low nitrogen conditions, nearly 82.1% and 17.9% showed P1-like (>59 cm) and P2-like (≤59 cm) culm length, respectively (Figure 2C). Additionally, of the 117 DH lines cultivated under normal nitrogen level, about 37.6% and 62.4% had long panicles (P1-like) and short panicles (P2-like), respectively (Figure 2D), while under low nitrogen cultivation, 39.3% and 60.7% had long panicles (P1-like) and short panicles (P2-like), respectively. Furthermore, from the set of the mapping population grown under normal range of nitrogen level, we observed that about 93.2% exhibited an increase in number of panicles per plant (P1-like), and 6.8% had P2-like number of panicles. Similarly, when grown under low nitrogen cultivation conditions, 98.3% and 1.7% of the DH lines recorded an increase in number of panicles per plant, P1-like and P2-like, respectively (Figure 2E).

We were also interested to see the changes in the root growth and the biomass dry weight of the mapping population under low nitrogen cultivation compared to the normal range nitrogen supply. Therefore, we measured the root length, shoot dry weight, and root dry weight. The results in Figure 2F indicate that only 3.4% of the DH lines had P1-like root growth pattern (relatively long roots), while nearly 96.6% of the same population grown under normal nitrogen cultivation exhibited a P2-like root growth pattern (relatively short roots). When grown under low nitrogen level, we observed that only 7.7% of the DH lines showed a P1-like root growth pattern against 92.3% with a P2-like root growth pattern (Figure 2F). Regarding the shoot dry weight of the mapping population cultivated under normal nitrogen conditions, our data indicate that about 96.6% and 3.4% had P1-like (high) and P2-like (low) shoot dry weight (SDW), while about 87.2% and 12.8% recorded a high SDW (P1-like) and low SDW (P2-like) under low nitrogen cultivation conditions (Figure 2G). As for root dry weight (RDW), about 81.2% and 14.5% recorded a high (P1-like) and low (P2-like) RDW under normal nitrogen cultivation, respectively; whereas, about 39.3% and 60.7% had a P1-like and P2-like RDW under low nitrogen cultivation (Figure 2H).

In the same perspective, the root–shoot ratio (R:S) results indicate that about 46.2% and 38.5% of the DH lines recorded a high R:S (P1-like, R:S > 0.42 and >0.35 based on length (Figure 2I) and dry weight (Figure 2J), respectively), while 61.5% and 53.8% (P2-like, R:S ≤ 0.42 and ≤0.35 based on length and dry weight, respectively) showed a low R:S (P2-like). Meanwhile, under low nitrogen conditions, about 19.7% and 62.4% (R:S based on length and dry weight, respectively) had a P1-like R:S (>0.35 and >0.37, based on length and dry weight, respectively) against 37.6% and 59% (R:S ≤ 0.37 and ≤0.35, based on length and dry weight, respectively). 

In addition, the phenotypic response of the parental lines (Figure 2A–L) indicates that both 93-11 (P1) and Milyang352 (P2) experienced a reduction in almost all the traits when grown under low nitrogen cultivation, compared to the normal nitrogen supply conditions, except a slight increase in culm length recorded in 93-11 (1.1% increase) and Milyang352 (0.4% increase), and a significant increase in number of panicles per plant observed in 93-11 (P1) against 16.6% reduction recorded in Milyang352 (P2) under low nitrogen cultivation.

### 3.2. Relatedness, Principal Component Analysis, and Correlation between Traits

In Figure 3A, our data show the KASP marker-based kinship matrix, also known as the co-ancestry or half relatedness, revealing the distribution of coefficients of co-ancestry of the mapping population. Additionally, the DH lines were grouped into two distinct clusters with regard to their number of tillers pattern (P1-type or P2-type) under low nitrogen cultivation (Figure 3B). The principal component analysis (PCA) proposes a correlation between traits under normal and low nitrogen cultivation conditions. Principal component 1 (PC1), 2 (PC2), 3 (PC3), and 4 (PC4) explained 27.3%, 19.5%, 15.2%, and 14.1% of the proportion of variance of the observed phenotypes, respectively, of the DH lines grown under normal nitrogen conditions. Similarly, under low nitrogen cultivation, PC1, PC2, PC3, and PC4 explained 27.3%, 19.1%, 15.6%, and 14.4% of the proportion of variance of the observed phenotypes.

Initially, we were interested in investigating a possible relationship between tiller number and other quantitative traits of interest recorded in the mapping population. For this reason, we performed a linear regression analysis. The results shown in Figure 3C,D suggest a strong positive correlation between tiller number and panicle number (R^2^ = 0.603*** and R^2^ = 0.560***, under normal and low nitrogen cultivation, respectively). In contrast, plant height showed a negative correlation with tiller number (R^2^ = −0.465*** and R^2^ = −0.279***, under normal and low nitrogen cultivation, respectively). However, culm length (R^2^ = 0.085^ns^ and R^2^ = 0.198^ns^), panicle length (R^2^ = −0.199* and R^2^ = −0.222^ns^), root length (R^2^ = 0.046^ns^ and R^2^ = 0.003^ns^), and root dry weight (R^2^ = 0.173^ns^ and R^2^ = 0.288**) showed a nonexistent or very low correlation with tiller number (Appendix A). In the same perspective, root length and root dry weight had a positive but weak correlation under normal nitrogen cultivation (R^2^ = 0.209*, Appendix A), while showing a stronger positive correlation under low nitrogen cultivation (R^2^ = 0.440***, Appendix A). Likewise, shoot dry weight and root dry weight are predicted to have a strong positive correlation. Furthermore, it was interesting to see that shoot dry weight (SDW) of the mapping population showed a strong positive correlation with tiller number (R^2^ = 0.583***, Figure 3F, Appendix A) under low nitrogen cultivation conditions, while under normal nitrogen supply, a weak positive correlation was observed (R^2^ = 0.317***, Figure 3E, Appendix A). It is then thought that a unit increase in the number of tillers would cause an increase in the shoot dry weight of rice by the same measure. Appendix A provide details information about the Person r correlation between traits. Data in these two tables also revealed that the correlation level between tiller number and other traits differ depending on each nitrogen condition.

### 3.3. Quantitative Trait Loci (QTLs) Associated with Plant Growth and Shoot Branching under Normal and Low Nitrogen Cultivation Were Mapped to Different Chromosomes

We used 240 KASP makers and Fluidigm markers, and the phenotype data of 117 DH lines and parental lines, cultivated under normal nitrogen and low nitrogen levels, to perform linkage mapping and QTL analysis. The results revealed that, in total, 28 QTLs (all traits considered) associated with plant growth and shoot branching were mapped to all rice chromosomes, except chromosomes 6 and 11 (Table 2, Figure 4A–J). In essence, five QTLs (*qPHN1-1*, LOD: 4.7; *qPHN3-1*, LOD: 4.88; *qPHN5-1*, LOD: 5.69; *qPHN10-1*, LOD: 5.3; and *qPHN12-1*, LOD: 5.13) were associated with plant height of rice plants grown under normal nitrogen cultivation (Table 2, Figure 4A,C,E,I,J). The additive effect recorded by *qPHN1* (2.08) and *qPHN10* (2.14), and those of the QTLs *qPHN3* (−2.1), *qPHN5* (−2.2), *qPHN12* (−2), indicate that the alleles from both 93-11 and Milyang352 contributed to the observed plant height phenotypic variation under normal nitrogen cultivation.

In the case of tiller number, two QTLs (*qTNN2-1*, LOD: 4.14; and *qTNN4-1*, LOD: 3.92) were detected in DH lines cultivated under normal nitrogen conditions, accounted for about 14.5% of the phenotypic variance explained (PVE) (Table 2, Figure 4B,D). However, solely one QTL (*qTNL2-1*, LOD:6.05) mapped to chromosome 2, flanked by KJ02_057 and id2012773, was associated with tiller number in DH lines grown under low nitrogen cultivation conditions, and accounted for about 20.5% of the PVE (Table 2, Figure 4B). The recorded additive effect (1.81) indicates that the allele from 93-11 (P1, *indica*) contributed to the observed phenotypic variation for tiller number under low nitrogen cultivation. 

In addition, seven QTLs associated with culm length, of which number five were detected in the mapping population cultivated under normal nitrogen conditions (*qCLN2-1*, LOD: 4.42; *qCLN2-2*, LOD: 13.45; *qCLN3-1*, LOD: 11.45; *qCLN9-1*, LOD:4.56; and *qCLN10-1*, LOD: 3.18) (Table 2, Figure 4B,C,H,I). The QTL *qCLN2-1* (for culm length) and the *qTNL2-2* (for tiller number) coincided within the same region and are flanked by the same markers (KJ02_057 and id20112773), which would suggest that *qTNL2-2* could be associated with both tiller number and culm length. Furthermore, two QTLs (*qCLL1-1*, LOD: 3.95; and *qCLL9-1*, LOD: 4.63) were detected in DH lines grown under low nitrogen level (Table 2, Figure 4A,H). The recorded additive effects indicate that both 93-11 and Milyang352 contributed to the observed culm length phenotypic variation.

In the same way, a set of five QTLs (*qPLN2-1*, LOD: 4.81; *qPLN3-1*, LOD: 3.92; *qPLN3-2*, LOD: 8.84; *qPLN7-1*, LOD: 5.94; and *qPLN8-1*, LOD: 8.06) associated with panicle length were only detected in the mapping population cultivated under normal nitrogen conditions (Table 2, Figure 4B,C,F,G). The additive effects recorded for both normal and low nitrogen cultivation conditions show that 93-11 and Milyang352, P1 and P2, respectively, contributed to the observed panicle length phenotypic variation. For panicle number trait, a total of four QTLs were detected, of which number three (*qPNN2-1*, LOD: 3.49; *qPNN4-1*, LOD: 9.63; and *qPNN8-1*, LOD: 4.34) were associated with normal nitrogen cultivation regime (Table 2, Figure 4B,D,G), and one QTL (*qPNL4-1*, LOD: 2.95) with low nitrogen cultivation (Table 2, Figure 4D). The additive effects indicate that, for both normal and low nitrogen cultivation conditions, the alleles from the *indica* parental line (93-11, P1) contributed to the observed phenotypic variation for panicle number (Table 3). 

A successful establishment of plants is expected when they develop a robust rooting system for efficient nutrients and water acquisition. Our results indicate that a single QTL (*qRLL4-1*, LOD:3.41) associated with root growth under low nitrogen cultivation was detected, and mapped to chromosome 4, and flanked by KJ04_053 and KJ04_057 (closest marker) KASP markers. The additive effect (−1.1) suggests that the allele from Milyang352 (P2, *japonica*) contributed to the observed phenotypic variance explained (PVE, 11.98%) (Table 2, Figure 4D). Biomass dry weight is affected by many factors, including the acquisition of mineral nutrients and crop productivity [51]. It was also proposed that there is an interdependent relationship between the shoot and roots [52,53]. The QTL analysis results showed that one QTL associated with shoot dry weight (*qSDWN5-1*, LOD: 3.74) (Table 2, Figure 4E), and two QTLs (*qRDW1-1*, LOD: 4.06; and *qRDW4-1*, LOD: 3.22) (Table 2, Figure 4A,D). 

Although many QTLs associated with different traits analyzed were identified, we opted to unveil candidate genes found within *qTNL2-1* (684.4 kbp, Chr2:26706424..27390847) essentially due to the fact that this major QTL was the only one associated with the control of shoot branching under low nitrogen cultivation (Table 3), which may help reduce excessive nitrogen-rich fertilizers application in rice cultivation, and contribute to reducing methane (NH_4_) emissions from paddy fields. An additional list of candidate genes with interesting annotated functions, pooled from the *qTNN4-1* region (1.193 Mbp) and associated with tiller number under normal nitrogen cultivation conditions, is attached as Appendix A.

Shoot branching has been shown to be regulated at different levels of the plant cellular metabolism. It is well established that phytohormones signaling pathways are key regulators of shoot branching in plants, operating in synergetic or antagonistic manner. Among the identified candidate genes (Table 3), we found two genes encoding GRAS (GIBBERELILIN-INSENSITIVE (GAI), REPRESSOR of GA1-3 (RGA), and SCARECROW (SCR)) transcription factor (TF) (Os02g44360 and Os2g44370). Proteins in the GRAS domain family are known as key players in gibberellin (GA) signaling, which regulates various aspects of plant growth and development, by inhibiting the proliferation and expansion of cell-mediated plant growth [54]. Another gene found in the same region is *OsGPX3*, a glutathione peroxidase putative expressed earlier reported to be involved shoot and root development in rice and stress response [55], as well as abscisic acid (ABA) response in rice [56]. Similarly, we could also find many other genes with interesting predicted functions, such as *OsDhn1* (Os02g44870), earlier suggested to be involved in abiotic stress response in rice [57,58], *OsRALF8* (rapid alkalinization factor, Os02g44940) gene, one of which FERONIA (a receptor-like kinase (RLK) member of CrRLK1L, known as a major plant cell growth modulator in distinct tissues) is known to be a receptor, and previously reported to play a critical role in the regulation of plant defense and development (inhibitory or suppression of growth) [59]. 

Reports indicated that carbohydrates regulate different aspects of plant growth via the modulation of cell division and expansion. For instance, a gene encoding DUF581 domain was reported to be a generic SnRK1 (sucrose nonfermenting-1 (SNF1)-related protein kinase) interaction module and co-expressed with SnRK1 during plant cell signaling [60]. We also identified a DUF encoding gene *OsFBDUF13* (F-box-like domain and domain of unknown function (DUF295) containing protein) within the *qTNL2-1* region. In the same way, a gene identified as being involved in carbohydrates metabolic process, among other biological processes, was found (*OsHUGT1*, Os02g44510). The other set of genes encoding the Aquaporin (*OsPIP1* (plasma membrane intrinsic protein 1) or *OsRWC1* (water channel protein), Os02g44630), the cytochrome P450 (Os02g44654) [61,62], zinc finger RING-type domain (Os02g44700), *OsSID1* or *OsIDD4* (Os02g45054) [63], and MYB (Myeloblastosis) transcription factor encoding gene (Os02g4580) was found within the detected QTL, among other genes (Table 3). 

## 4. Discussion

### 4.1. Tiller Number Positively Correlated with Shoot Dry Weight, While Shoot and Root Dry Weights Showed Positive Relationship

In rice, tillers represent the branches that develop from the leaf axils at each unelongated node of the main shoot or from other tillers during vegetative growth. It is said that tiller number determines the number panicles per plant [39,64,65], which happens to be an important yield component of rice. The number of tillers is influenced by environmental factors [66], agricultural practices, including nitrogen fertilization [67,68,69,70,71,72], or genetic factors [65,73,74,75,76,77]. It is also expected that an increase or decrease in tiller number per plant could cause significant changes in the shoot biomass. Here, the results of the linear regression analysis showed that shoot dry weight positively correlated with tiller number (R^2^ = 0.583*** and R^2^ = 0.317***, under low and normal nitrogen cultivation, respectively) and plant height (R^2^ = 0.374*** and R^2^ = 0.350***, under low and normal nitrogen cultivation, respectively). Therefore, these findings suggest that a unit increase in tiller number would result in the increase in shoot dry weight by the same measure. In the same way, we recorded that panicle number showed a strong positive correlation with tiller number (R^2^ = 0.560*** and R^2^ = 0.603***, under low and normal nitrogen cultivation, respectively). It could then be said that rice plants with a high tiller number would eventually develop more panicles, which would result in an increased productivity. 

Roots and shoot are said to be positionally and developmentally related in various ways [78,79]. Therefore, the recorded positive correlation between shoot dry weight and root dry weight (R^2^ = 0.491*** and R^2^ = 0.309***, under low and normal nitrogen cultivation, respectively), coupled with the respective phenotypes of the mapping population concerning SDW and RDW (Figure 2G,H: 96.6% and 87.2% of the DH lines showed a high SDW and RDW, respectively, similar to 93-11 (P1, *indica* and high tillering cultivar)), would suggest that a robust rooting system may contribute to achieving an optimum shoot growth and development in rice. In addition, it is thought that the observed difference of the correlation level between tiller number and other traits under normal and low nitrogen conditions would also give insights into the recorded N-effects on the phenotypic traits analyzed, and on the genetic loci difference controlling tiller number.

### 4.2. Quantiative Trait Loci (QTLs) Associated with Shoot Branching under Low Nitrogen Cultivation Conditions Are Identified

Several agronomic traits of rice have been suggested to be controlled by multiple quantitative trait loci (QTLs) [27,80,81]. A number of studies have proposed a large number of putative QTLs controlling tiller number and panicle number in rice. Most of the reported QTLs related to tiller number or panicle number are mapped across the 10–11 chromosomes of rice, using various methods, marker-types on different sorts of rice populations, at various developmental stages [27,39,82]. In this study, of the 28 QTLs identified for all the growth related and productivity related traits, only three were associated with tiller number. Of this number, two (*qTNN2-1* and *qTNN4-1*) were detected in the set of DH lines grown under normal nitrogen conditions, and one (*qTNL2-1*) in that grown under low nitrogen conditions, were mapped on chromosomes 2 and 4. Meanwhile, four QTLs related to panicle number were detected (Table 3), with *qPNL4-1* being the only QTL associated with panicle number identified in DH lines grown under low nitrogen conditions, and mapped on chromosome 4. A study conducted by Liu et al. [82] identified 12 QTLs associated with tiller number mapped on chromosomes 2 (two QTLs: 12.5 cM and 23.1 cM), 3 (three QTLs: 6.0 cM, 17.2 cM, and 20.4 cM) and 6–9 (three QTLs on chromosome 6: 8.5 cM, 8.8 cM, and 28 cM; one on chromosome 7: 24.5 cM; one on chromosome 8: 8.3 cM; one on chromosome 9: 17.0 cM, and one on chromosome 12: 2.6 cM) using rice microsatellites (RM) markers on single segment substitution lines (SSSLs). In a similar perspective, Yan et al. [39] reported 15 QTLs associated with tiller number at different growth stages, and mapped on chromosomes 1–4, 6–8, and 12. In a converse approach, Liu et al. [27] identified 27 QTLs associated with tiller number at different development stages in rice, and mapped on chromosomes 1–12. However, Zhu and his colleagues reported a QTL associated with panicle number in rice, mapped on chromosome 1 (*qPN1*) [83]. None of the previously reported QTLs associated with tiller number or panicle number overlapped with the identified QTLs by this study. However, some of them are mapped on similar chromosomes but different positions.

### 4.3. QTLs qTNL2-1 and qTN4-1 Harbor Candidate Genes Proposed to Be Involved in the Control of Shoot Branching in Rice under Low and Regular Nitrogen Cultivation

In the perspective of unveiling the identity of candidate genes located within the genetic region covered by the QTL *qTNL2-1* associated with the control of shoot branching under low nitrogen cultivation, we used the available maker positions to explore this region. Interestingly, a substantial number of genes were proposed to have transcription factor binding activity. This is the case for two GRAS transcription factor (TF) encoding genes (Os02g44360 and Os2g44370), which have a sequence similarity of about 86.6%. Proteins in the GRAS domain family are known as key players in gibberellin (GA) signaling [84,85], which regulates various aspects of plant growth and development, by inhibiting the proliferation and expansion of cell-mediated plant growth [54]. Similarly, *OsHMG1* (Os02g44930), *OsSID1* (Os02g45054), also known as *OsIDD4* (a Zinc finger C_2_H_2_-type domain containing protein), and MYB TF encoding gene (Os02g45080), are proposed to code for transcription factors. 

Another set of interesting genes, such as *OsGPX3,* was earlier reported to be involved in the development of the roots and shoot in rice, in the stress response mechanisms [55], and abscisic acid (ABA) response in rice [56]; *OsDhn1*, earlier suggested to be involved in abiotic stress response in rice [57,58]; *OsRALF8* [86], previously reported to play a critical role in the regulation of plant defense and development (inhibition or suppression of growth) [59], could play important roles in the control of shoot branching, while maintaining a balanced plant growth and development under low nitrogen cultivation in rice. Nietzsche et al. [60] supported that carbohydrates regulate different aspects of plant growth via the modulation of cell division and expansion. Here, a DUF encoding gene *OsFBDUF1* or *OsMAIF23* found within the *qTNL2-1* region, and *OsHUGT1* known as UDP-glucose glycoprotein glucosyltransferase 1 precursor, predicted as being involved in carbohydrates metabolic process would contribute to the observed tiller number phenotype under low nitrogen cultivation. 

In the same way, Aquaporin (*OsPIP1* or *OsRWC1*), and *OsDhn1* or *OsLip9* [57,58], associated with cytochrome P450 (earlier suggested to be involved in auxin signaling by Chaban et al. [87] or encoded by *OsMAX1*, a strigolactone biosynthesis genes according to Marek et al. [61,88] or encoded by *OsSLR1/OsGAI* (*SLENDER RICE 1* encodes a DELLA protein belonging to GRAS [89]), which modulate gibberellin response in rice [62]), are believed to actively contribute to modulating water movements in and outside the cell to facilitate transport of nutrients, including nitrogen, as well as plant growth and development under low nitrogen conditions. From another perspective, Zou et al. [90,91] identified and characterized the *OsHTD1* (*HIGH-TILLERING DWARF1,* Os04g46470) gene encoding a carotenoid cleavage dioxygenase (CCD7 or D7 ortholog of *Arabidopsis MAX3*, involved in the strigolactone biosynthesis). The *OsHTD1* was reported to be required for negative regulation of the outgrowth of axillary buds. Similarly, Wang et al. [29] demonstrated that a partial loss-of-function allele of *OsHTD1*/CCD7 gene increased tiller number and improved rice grain yield. Here, our finding revealed that the *OsHTD1* gene (Chr4:27567824..27570926) is located within the QTL *qPNL4-1* mapped on chromosome 4, identified by this study as being associated with the control of panicle number under low nitrogen cultivation. The *qPNL4-1* is flanked by ad04009559 (left marker, 26443283 kbp) and ah04001252 (right marker, 28788134 kbp).

The breakthrough is that many of the candidate genes found either in *qTNL2-1* or *qTNN4-1* are also proposed to be involved in plant stress signaling or response mechanisms. A recent study investigating the transcriptional regulation of hormonal biosynthesis or signaling pathways genes under drought stress conditions, proposed that shoot branching related genes could be involved in the adaptive response mechanisms of plants towards abiotic stress tolerance in *Arabidopsis* [92]. We could speculate that under normal growth conditions, plants utilize their resources for their growth and development. However, under a reduced level of nitrogen, which could also be perceived as a stress, plants would activate abiotic stress responsive genes and/or growth related genes, in order to either escape or avoid the stress or alleviate the adverse effects that would be caused by the stress, while tending to maintain a balanced growth. Owing to the above, a possible crosstalk between plant growth and development and defense mechanisms could be suggested.

## 5. Conclusions

Shoot branching is an important agronomic trait contributing to the yield and productivity of cereal food crops, such as rice. In plants, shoot branching is controlled by a complex hormonal signaling network, operating either in synergetic or antagonistic manner. It is well established that most of the genes involved in the regulation of shoot branching, through promotion or inhibition, are also involved in the adaptive response mechanisms of plants towards stress tolerance or resistance (abiotic or biotic stress). In this study, we identified QTLs proposed to be associated with tiller number and panicle number under low nitrogen cultivation, mapped to chromosomes 2 (*qTNL2-1*) and chromosome 4 (*qPNL4-1*). Another QTL associated with tiller number was detected in the doubled haploid population grown under normal nitrogen cultivation conditions (*qTNN4-1*) and mapped on chromosome 4. These QTLs harbor candidate genes with interesting annotated functions, and proposed for playing important roles in the adaptive response mechanisms towards stress (abiotic and biotic) in rice, in addition to being associated with shoot branching. Candidate genes encoding GRAS protein, cytochrome P450, *OsRALF8*, Aquaporin and *OsDhn1*, among others, would play important roles in controlling shoot branching under low nitrogen cultivation with regard to their functions and their similarity with previously reported genes. Therefore, this study suggests a possible crosstalk between the regulation of plant growth and development, and the stress response mechanisms. Functional analysis would help identify major genes and elucidate their function.

## Figures and Tables

**Figure 1 genes-12-00745-f001:**
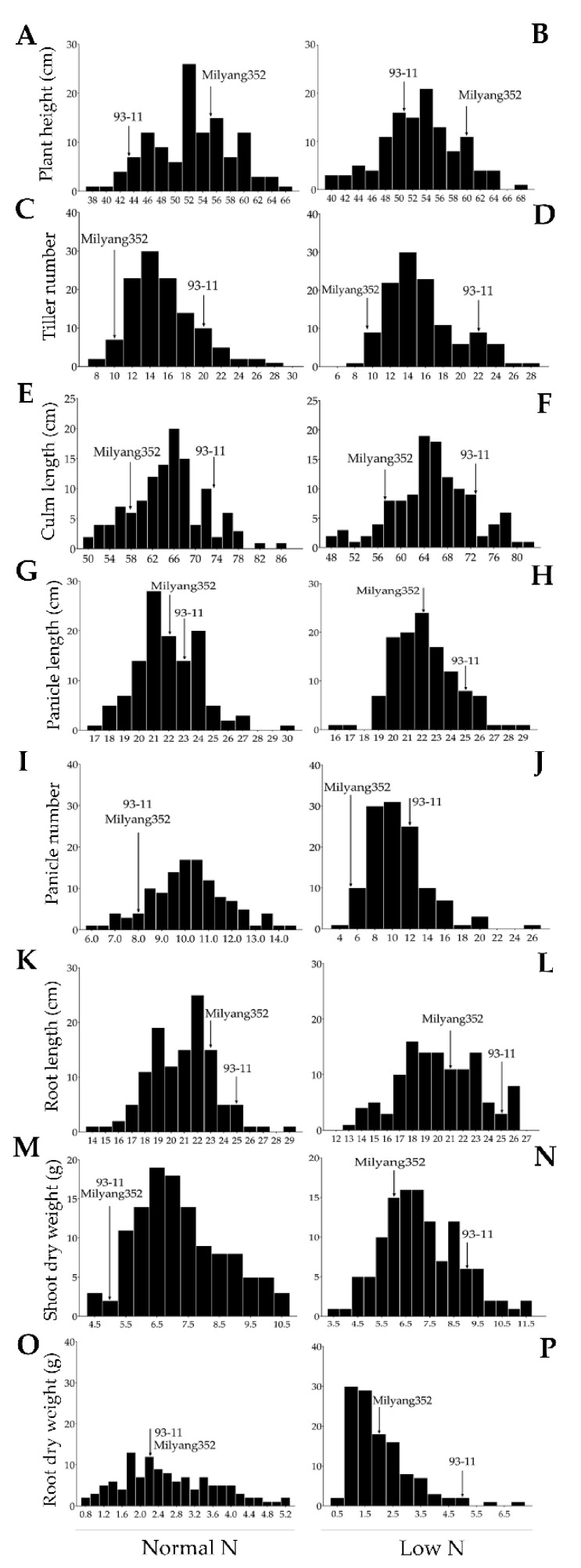
Frequency distribution of traits. Frequency distribution of plant height of a DH population grown under a normal (**A**) and low (**B**) nitrogen (N) cultivation regimes. Frequency distribution of tiller number under normal (**C**) and low (**D**) nitrogen levels. Frequency distribution of culm length (**E**,**F**) and panicle length (**G**,**H**) under normal and low N cultivation. Frequency distribution of panicle number (**I**,**J**) and root length (**K**,**L**) under normal and low N cultivation. Frequency distribution of shoot (**M**,**N**) and root (**O**,**P**) dry weights under normal and low N levels.

**Figure 2 genes-12-00745-f002:**
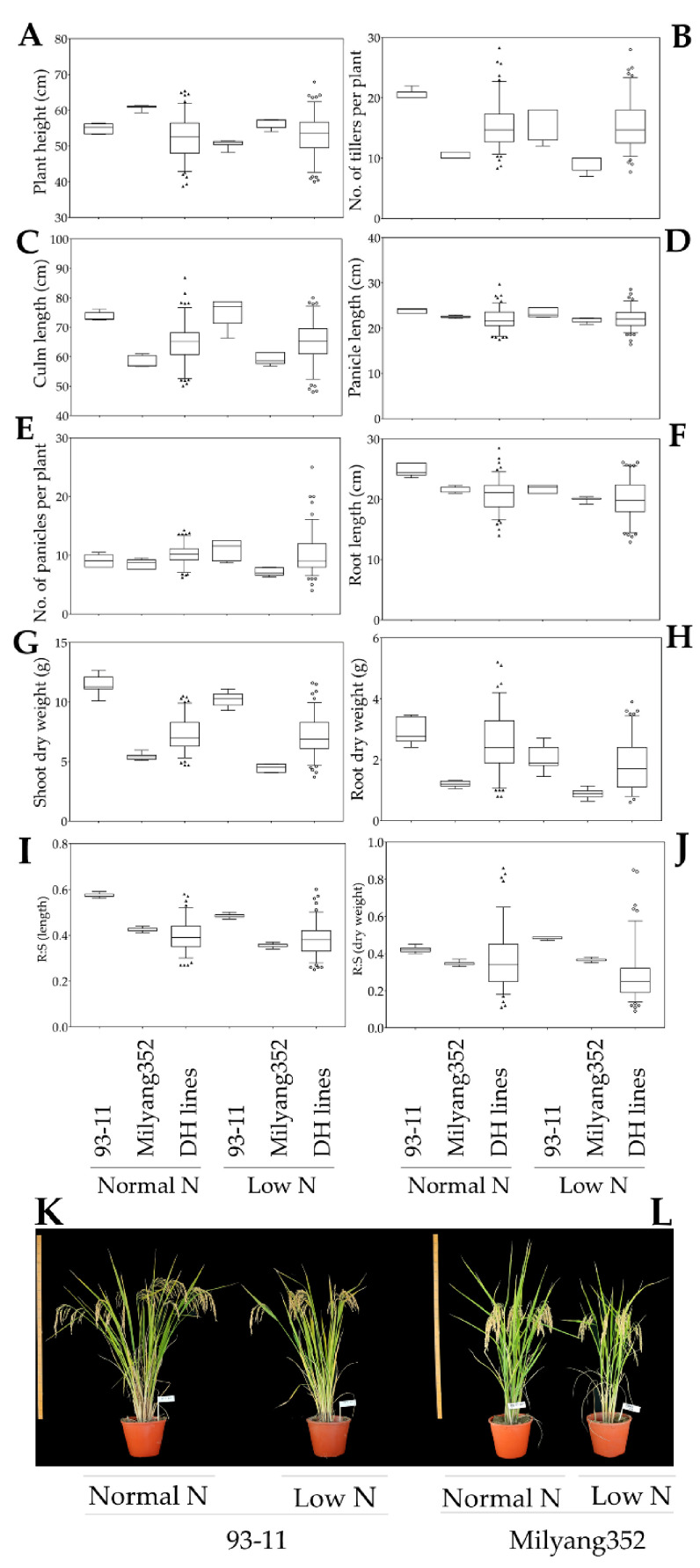
Genotype–phenotype correlation of the doubled haploid population and parental lines under normal and low nitrogen cultivation conditions. (**A**) The box plot displays the effects of a low nitrogen (N) cultivation regime on the growth of 93-11 (P1, indica), Milyang352 (P2, japonica), and that of a doubled haploid (DH) rice population (*n* = 117) compared to the normal N application, (**B**) box plot showing the number of tillers per plant, (**C**) culm length, (**D**) panicle length, (**E**) number of panicles per plant, (**F**) root length, (**G**) shoot dry weight, (**H**) root dry weight, (**I**) root–shoot (R:S) ratio calculated from the root length and plant height, and (**J**) based on the root dry weight and shoot dry weight of the same population, and (**K**,**L**) phenotypes of 93-11 (P1) and Milyang352 (P2) under the same conditions.

**Figure 3 genes-12-00745-f003:**
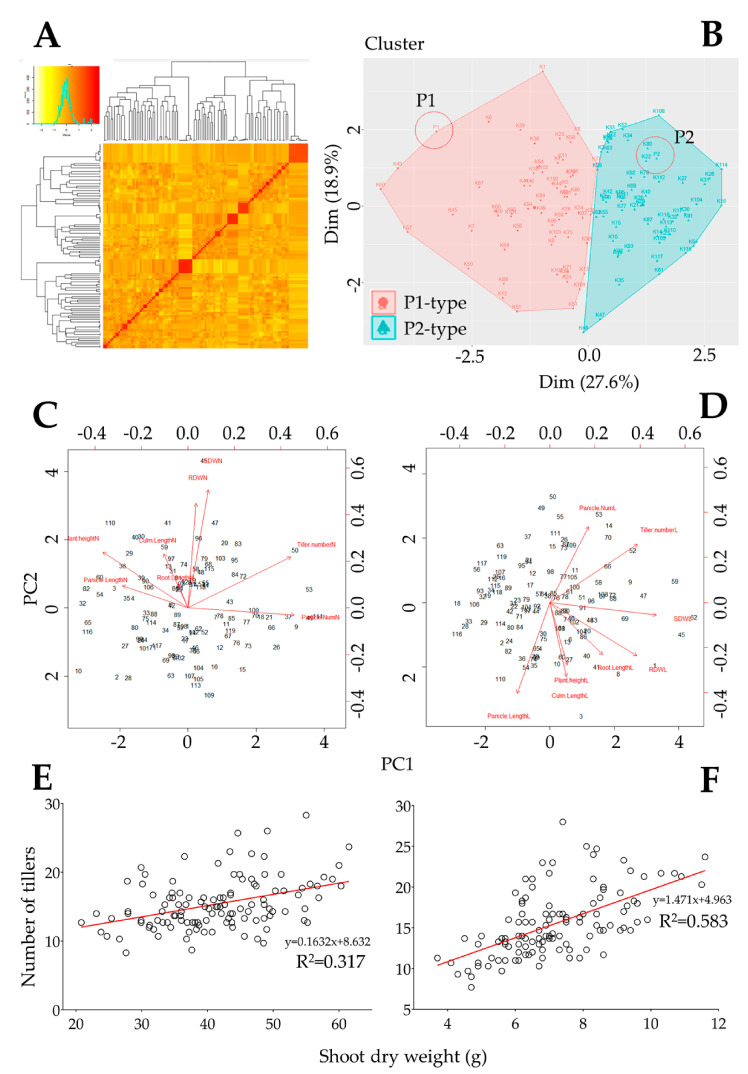
Population genome background and principal component analysis results. (**A**) The density map of pairwise kinship matrix values generated based on 240 KASP markers on a DH population, (**B**) cluster plot showing the distinctive phenotypic response between P1 (93-11) and P2 (Milyang352) based on the number of tillers under low N cultivation, (**C**,**D**) 2D principal component analysis (PCA) indicating the relationship between traits, and (**E**,**F**) linear regression analysis proposing the existence of a positive correlation between number of tillers and shoot dry weight in rice grown under a normal and low N cultivation regime, respectively.

**Figure 4 genes-12-00745-f004:**
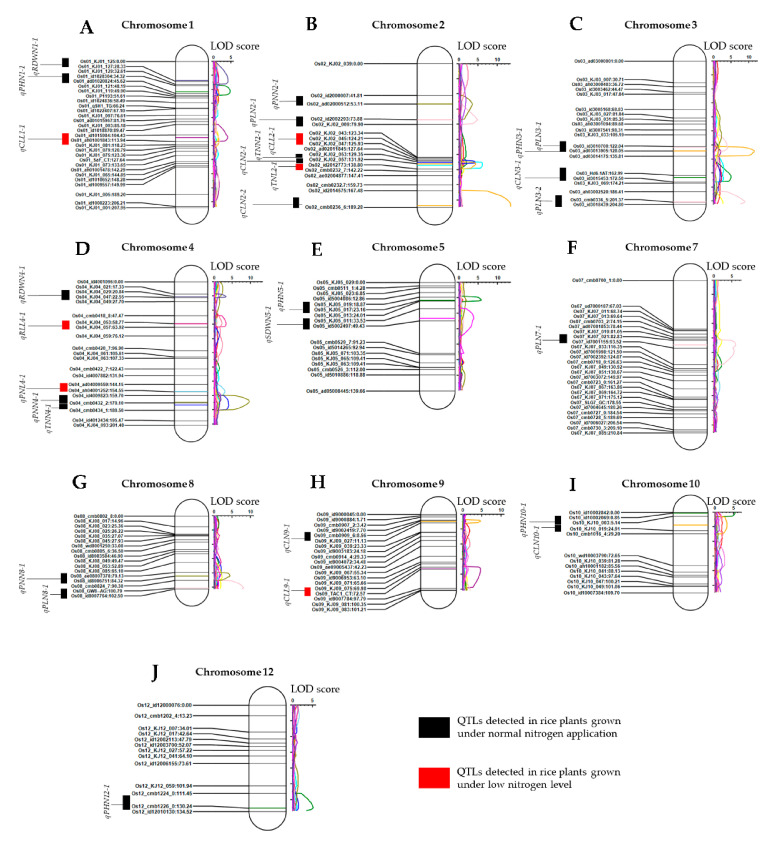
Linkage maps and identified QTLs associated with various agronomic traits in rice under normal and low nitrogen cultivation. Five (5) QTLs (*qPHN-1*, 43 cM; *qPHN3-1*, 166 cM; *qPHN5-1*, 23 cM; *qPHN10-1*, 1 cM; and *qPHN12-1*, 130 cM) associated with plant height of rice were mapped on five (5) different chromosomes, for normal nitrogen cultivation. The QTLs associated with tiller number were mapped on chromosome 2 (*qTNN-1*, 130 cM for normal nitrogen cultivation, and *qTNL2-1*, 134 cM for low nitrogen cultivation) and chromosome 4 (*qTNN4-1*, 173 cM). For both normal and low nitrogen cultivation conditions, five QTLs (*qCLN2-1*, 132 cM; *qCLN2-2*, 187 cM; *qCLN3-1*, 128 cM; *qCLN9-1*, 9 cM; and *qCLN10-1*, 17 cM) and two (2) QTLs (*qCLL1-1*, 108 cM; and *qCLL9-1*, 72 cM) were mapped on four and two chromosomes, respectively. Five QTLs (*qPLN2-1*, 74 cM; *qPLN3-1,* 123 cM; *qPLN3-2*, 201 cM; *qPLN7-1*, 89 cM; and *qPLN8-1,* 101 cM) associated with panicle length under normal nitrogen cultivation were mapped on four chromosomes. Three QTLs (*qPNN2-1*, 53 cM; *qPNN4-1*, 169 cM; *qPNN8-1*, 84 cM) and one QTL (*qPNL4-1*, 154 cM) associated with panicle number were mapped on three chromosomes and one chromosome, for normal nitrogen and low nitrogen cultivation, respectively. A unique QTL (*qRLL4-1*, 59 cM) associated with root length was mapped on chromosome 4. Finally, one QTL (*qSDWN-51*, 46 cM) and two QTLs (*qRDW1-1*, 27 cM; and *qRDW4-1*, 21 cM) associated with shoot dry weight (SDW) and root dry weight, respectively, under normal nitrogen cultivation, were mapped to chromosomes 5 (for SDW) and 1 and 4 (for RDW). (**A**–**J**) Chromosomes 1, 2, 3, 4, 5, 7, 8, 9, 10, and 12. LOD score, logarithm of the odds indicating the significance of genetic linkage between markers and the detected QTLs above a threshold.

**Table 1 genes-12-00745-t001:** Soil mineral elements composition and hydrogen potential.

Field	pH	TN (%)	EC	OM (%)	P_2_O_5_ (mg kg^−1^)	K^+^ (cmolc kg^−1^)	Ca^2+^ (cmolc kg^−1^)	Mg^2+^ (cmolc kg^−1^)	Na^+^ (cmolc kg^−1^)
Low N	5.5	0.27	0.75	2.74	44.77	0.84	4.76	1.01	0.18
Normal N	5.7	1.38	1.38	3.79	96.87	0.27	1.17	7.82	1.27

pH: potential of hydrogen; TN: total nitrogen; EC: cationic exchange; OM: organic matter; P_2_O_5_: phosphate; K^+^: potassium; Ca^2+^: calcium; Mg^2+^: magnesium; Na^+^: sodium, cmolc kg^−1^: centimoles of charges per kilogram.

**Table 2 genes-12-00745-t002:** Detected QTLs associated with growth and development, and yield components of rice under normal and low cultivation.

Traits (a)	QTL (b)	Chr.(c)	Position (cM) (d)	Left Marker (e)	Right Marker (f)	LOD (g)	PVE (%) (h)	Add (i)
PHN	*qPHN1-1*	1	43	Os01_id1028304	Os01_ad01020824	4.703	8.745	2.08
	*qPHN3-1*	3	166	Os03_Hd6-1AT	Os03_id3015453	4.883	8.798	−2.1
	*qPHN5-1*	5	23	Os05_KJ05_019	Os05_KJ05_017	5.694	9.915	−2.2
	*qPHN10-1*	10	1	Os10_id10002069	Os10_KJ10_003	5.305	9.147	2.14
	*qPHN12-1*	12	130	Os12_cmb1224_0	Os12_cmb1226_0	5.134	8.836	−2
TNL	*qTNL2-1*	2	134	Os02_KJ02_057	Os02_id2012773	6.052	20.45	1.81
TNN	*qTNN2-1*	2	130	Os02_KJ02_053	Os02_KJ02_057	4.141	14.51	1.25
	*qTNN4-1*	4	173	Os04_cmb0432_2	Os04_cmb0434_1	3.922	14.49	1.33
CLL	*qCLL1-1*	1	108	Os01_id1015984	Os01_ah01001843	3.947	10.57	3.3
	*qCLL9-1*	9	72	Os09_KJ09_075	Os09_TAC1_CT	4.632	11.59	−2.8
CLN	*qCLN2-1*	2	132	Os02_KJ02_057	Os02_id2012773	4.42	6.509	−1.9
	*qCLN2-2*	2	187	Os02_id2014575	Os02_cmb0236_6	13.45	24.81	3.77
	*qCLN3-1*	3	128	Os03_id3010700	Os03_ad03013905	11.45	19.51	3.37
	*qCLN9-1*	9	9	Os09_cmb0909_6	Os09_KJ09_027	4.556	6.805	−2.2
	*qCLN10-1*	10	17	Os10_KJ10_003	Os10_KJ10_019	3.178	5.401	1.89
PLN	*qPLN2-1*	2	74	Os02_id2002293	Os02_KJ02_009	4.811	7.985	−0.7
	*qPLN3-1*	3	123	Os03_id3010700	Os03_ad03013905	3.917	6.768	0.61
	*qPLN3-2*	3	201	Os03_ah03002520	Os03_cmb0336_5	8.844	16.06	0.93
	*qPLN7-1*	7	89	Os07_KJ07_021	Os07_id7001155	5.938	11.14	0.78
	*qPLN8-1*	8	101	Os08_GW8-AG	Os08_id8007764	8.059	14.35	−0.9
PNL	*qPNL4-1*	4	154	Os04_ad04009559	Os04_ah04001252	2.954	11.99	1.18
PNN	*qPNN2-1*	2	53	Os02_id2000007	Os02_ad02000512	3.487	7.269	0.63
	*qPNN4-1*	4	169	Os04_id4009823	Os04_cmb0432_2	9.633	23.76	0.94
	*qPNN8-1*	8	84	Os08_ae08007378	Os08_id8006751	4.34	9.338	0.55
RLL	*qRLL4-1*	4	59	Os04_KJ04_053	Os04_KJ04_057	3.241	11.99	−1.1
SDWN	*qSDWN5-1*	5	46	Os05_KJ05_011	Os05_id5002497	3.748	12.88	−0.6
RDWN	*qRDWN1-1*	1	27	Os01_KJ01_125	Os01_KJ01_127	4.069	11.74	−0.4
	*qRDWN4-1*	4	21	Os04_KJ04_029	Os04_KJ04_047	3.218	8.654	−0.3

**Table 3 genes-12-00745-t003:** Candidate genes pooled from the *qTNL2-1* associated with tiller number under low nitrogen availability.

No.	Gene Name	Locus	Annotation	Molecular Function	Cellular Component
1	*OsPBF19*	Os02g44220	Peroxisomal biogenesis factor 19, putative, expressed	Protein binding	Cytosol; peroxisome
2	*OsSc11/*	Os02g44360	Scarecrow transcription factor family protein, putative, expressed; GRAS: GIBBERELLIN-INSENSITIVE (GAI), REPRESSOR of GA1-3 (RGA), and SCARECROW (SCR)	Sequence-specific DNA binding transcription factor activity	Plastid, nucleus
3	*OsGRAS*	Os02g44370	Scarecrow, putative, expressed; GRAS transcription factor domain containing protein NSENSITIVE (GAI), REPRESSOR of GA1-3 (RGA), and SCARECROW (SCR)	Sequence-specific DNA binding transcription factor activity	Plastid, nucleus
4	*OsGpx3*	Os02g44500	Glutathione peroxidase, putative, expressed; OsGpx3|GPX3|OsGPx03	Glutathione peroxidase activity; peroxidase activity; oxidoreductase activity	Mitochondrion; plasma membrane; plastid; cytosol; chloroplast;
5	*OsHUGT1*	Os02g44510	UDP-glucose glycoprotein glucosyltransferase 1 precursor, putative, expressed	UDP-glucose:glycoprotein glucosyltransferase activity	Endoplasmic reticulum
6	*OsSub19*	Os02g44520	Putative Subtilisin homologue, expressed; Peptidase S8 and S53, subtilisin, kexin, sedolisin domain containing protein	Hydrolase activity; serine-type endopeptidase activity	Vacuole; cytoplasm; ribosome; cytosol; plastid; vacuolar membrane; chloroplast
7	*OsMalic*	Os02g44550	NADP-dependent malic enzyme, putative, expressed	Catalytic activity; malic enzyme activity; oxidoreductase activity, acting on the CH-OH group of donors	Cytosol
8		Os02g44570	Mitochondrial carrier protein, putative, expressed; Mitochondrial substrate carrier family protein	binding	Mitochondrial inner membrane
9	*OsPIP1;1|RWC1*	Os02g44630	Aquaporin. Plasma membrane intrinsic proteins subfamily	Transporter activity	Plasma membrane; mitochondrion; chloroplast envelop
10	*MAP3K.10—STE*	Os02g44642	STE_MEKK_ste11_MAP3K.10—STE kinases include homologs to sterile 7, sterile 11 and sterile 20 from yeast, expressed; Similar to MAP3Ka	Transferring phosphorus-containing groups; ATP binding; protein kinase activity	
11	*CYP450*	Os02g44654	Cytochrome P450, putative, expressed; Similar to Cytochrome P450	Heme binding; oxidoreductase activity	
12		Os02g44700	Zinc finger, C3HC4 type domain containing protein, expressed; Zinc finger, RING-type domain containing protein	Binding; catalytic activity; zinc ion binding; ubiquitin–protein transferase activity; metal ion binding; protein binding	
13	*OsDhn1/OsLip9*	Os02g44870	Dehydrin, putative, expressed; Dehydrin family protein	Binding	Cytosol; plasma membrane; plastid
14	*OsHMG1*	Os02g44930	HMG1/2, putative, expressed; Similar to HMGc1 protein	Sequence-specific DNA binding transcription factor activity; structural molecule activity	Intracellular
15	*OsRALFL8*	Os02g44940	Rapid Alkalinization Factor RALF family protein precursor, expressed; Rapid Alkalinization Factor family protein	Signal transducer activity	Extracellular region
15		Os02g44980	Transmembrane amino acid transporter protein, putative, expressed; Similar to amino acid transport protein	Transporter activity	Membrane; integral component of membrane
16	*OsFBDUF1/OsMAIF13*	Os02g44990	F-box and DUF domain containing protein, expressed; Cyclin-like F-box domain containing protein.	Protein binding	
17	bHLH	Os02g45010	Ethylene-responsive protein related, putative, expressed; Basic helix-loop-helix dimerization region bHLH domain containing protein	Sequence-specific DNA binding transcription factor activity; protein dimerization activity	Nucleus
18	*SID1|OsIDD4*	Os02g45054	ZOS2-15—C2H2-type domain containing protein, expressed; Zinc finger	Nucleic acid and protein binding; sequence-specific DNA binding transcription factor activity; metal ion binding	Nucleus
19	*MYB*	Os02g45080	MYB family transcription factor, putative, expressed, containing a myb_SHAQKYF class DNA-binding domain at the C-terminal end of the motif	DNA binding; sequence-specific DNA binding transcription factor activity	
20		Os02g45130	Protein kinase, putative, expressed	protein serine/threonine kinase activity; ATP binding	Nucleus
21	*OsALMT7*	Os02g45160	Aluminum-activated malate transporter, putative, expressed; Similar to ALMT3	Malate transport	

## Data Availability

Not applicable.

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
