# Peer review of "Novel QTL Associated with Shoot Branching Identified in Doubled Haploid Rice (Oryza sativa L.) under Low Nitrogen Cultivation"

_genes, 2021, doi:10.3390/genes12050745_

Round 1

Reviewer 1 Report

In the revised version of their work, authors provided a detailed responses to the concerns raised in the first version of the manuscript. Many aspects of the manuscript improved, yet there are some points that are not being fully addressed. These are:

Figure 4: The LOD score continues to be unexplained in the legend

Figure 1 Normality test: It is unclear to me why a normality test (e.g. Shapiro Wilk) cannot be done. It is also unclear which test was use to generate the p-values in Table S3. It is this the normality test? To claim normal distribution an appropriate test should be performed and explained in method.

In figures, it is still incorrect the location of the letter indicating the name of the panels, which should be placed in the same location in all panels (possibly top-left outside the graph area) to avoid confusion.

Please ensure that figure legends are updated to the new figure format (e.g. “–log10(p)” in figure 1 legend is not displayed in the figure panels).

Author Response

Title: Novel QTL Associated with Shoot Branching Identified in Doubled Haploid Rice (Oryza sativa L.) under Low Nitrogen Cultivation

Manuscript ID genes-1199924

Journal Genes

Section Plant Genetics and Genomics

Special Issue Genetic Diversity in Rice Cultivars

Point by point reply to reviewer’s comments

Reviewer 1

In the revised version of their work, authors provided a detailed responses to the concerns raised in the first version of the manuscript. Many aspects of the manuscript improved, yet there are some points that are not being fully addressed. These are:

The authors would like to thank the reviewer for his valuable comments and suggestions, which have helped us improve significantly the manuscript.

Figure 4: The LOD score continues to be unexplained in the legend

The authors apologize for the inconvenience. We have explained LOD score as recommended (line 431-432).

Figure 1 Normality test: It is unclear to me why a normality test (e.g. Shapiro Wilk) cannot be done. It is also unclear which test was use to generate the p-values in Table S3. It is this the normality test? To claim normal distribution an appropriate test should be performed and explained in method.

We are thankful to the reviewer for the concern raised. The frequency distribution was generated using “Frequency distribution function” in GraphPad Prism 7.0. Table S3 was generated during QTL analysis in IciMapping software, and the results and the Shapiro Wilk (W-statistic for the test of normality) was performed. And the p-value is that of W-test of normality. We have included the required explanation in lines 152–154.

In figures, it is still incorrect the location of the letter indicating the name of the panels, which should be placed in the same location in all panels (possibly top-left outside the graph area) to avoid confusion.

We apologize for the inconvenience. We have made the necessary changes as recommended (Figure 1–4, Figure S2)

Please ensure that figure legends are updated to the new figure format (e.g. “–log10(p)” in figure 1 legend is not displayed in the figure panels).

The authors would like to thank the reviewer for the concern. We think that in figure 1 we do not need to indicate the –log10(p), because this is the frequency distribution. We have removed this information related to –log10(p) form the Figure 1 caption. The –log10(p) is related to the quantile-quantile plot (Figure S2)

Reviewer 2 Report

Dear Authors,

Presented for review manuscript entitled: "Novel QTL Associated with Shoot Branching Identified in Doubled Haploid Rice (Oryza sativa L.) under Low Nitrogen Cultivation" showed very relationships between occurrence of QTLs with shoot branching. Generally I' recommend the manuscript for publication in Genes after major revision. The introduction contains to much information that are not related to main stream of manuscript. Only last 3 paragraph of introduction contains the valid information related to the paper. Moreover I have got the manuscript in which parts of the text were highlighted in green, why? Additionally the figure 1 should be much more clearly presented. I suggest to do it in the same style like figure 2 upper panel. I mean the Low and normal nitrogen labeling.

Author Response

Title: Novel QTL Associated with Shoot Branching Identified in Doubled Haploid Rice (Oryza sativa L.) under Low Nitrogen Cultivation

Manuscript ID genes-1199924

Journal Genes

Section Plant Genetics and Genomics

Special Issue Genetic Diversity in Rice Cultivars

Point by point reply to reviewer’s comments

Reviewer 2

Dear Authors,

Presented for review manuscript entitled: "Novel QTL Associated with Shoot Branching Identified in Doubled Haploid Rice (Oryza sativa L.) under Low Nitrogen Cultivation" showed very relationships between occurrence of QTLs with shoot branching. Generally, I' recommend the manuscript for publication in Genes after major revision.

The authors would like to thank the reviewer for his valuable comments and suggestions that have contributed to improving the content of our manuscript.

The introduction contains to much information that are not related to main stream of manuscript. Only last 3 paragraph of introduction contains the valid information related to the paper.

The authors appreciate the concern raised by the reviewer. We have tried to remove some information from the introduction section as suggested. However, we would like to specify that we found necessary to put into a broad context this study, regarding the spirit of the study. We intended to cover, in addition to the identification of genetic loci associated with shoot branching under different nitrogen cultivation regimes, subjects related to the use of nitrogen-rich fertilizers in rice production, which was shown to contribute to the emissions of greenhouse gases. A reduced application of nitrogen (low nitrogen level) using rice cultivars adapted to low nitrogen cultivation, may help reduce excessive application of nitrogen, while maintaining a balanced plant growth and development, and productivity.

Moreover, I have got the manuscript in which parts of the text were highlighted in green, why?

The authors appreciated the concern raised by the reviewer. This manuscript has been earlier reviewed by other reviewer, and the current version is a revised version of the manuscript. Which is why we highlighted green to indicate changes made for further processing.

Additionally, the figure 1 should be much more clearly presented. I suggest to do it in the same style like figure 2 upper panel. I mean the Low and normal nitrogen labeling.

The authors are thankful to the reviewer for his suggestions. We have modified the Figure 1 accordingly.

Round 2

Reviewer 2 Report

Dear authors thank you for taking my comments into account and including them in the manuscript.

This manuscript is a resubmission of an earlier submission. The following is a list of the peer review reports and author responses from that submission.

Round 1

Reviewer 1 Report

In this work Kwon and collaborators collected several developmental ad phenotypical parameters of 117 double haploid Oryza sativa rice lines original obtained by anther culture of the cross between varieties two main rice subspecies japonica and indica. The rice lines grown in field conditions with normal or low nitrogen input. Then, authors used KASP markers to map QTLs associated to specific traits for both nitrogen regimes used, and discussed the possible implication of their finding in the context of shoot branching. In general, mapping new QTLs in rice provides incremental knowledge toward the characterization of genes responsible for agronomical important rice traits, and therefore these results are of interest for the community. I however found this work rather descriptive, without real attempts to identify possible genes involved besides simply listing them. In addition, it is not well explained why authors are focussing only on few QTLs in the last part of the manuscript, and text and figures could be made clearer to the reader. Also, the title, the abstract and most of discussion are focusing quite strongly on shoot branching. However, most of result are about measurements of several traits and description of associated QTLs. There is not a real justification of why authors end focussing only on QTLs associated to shoot branching.  Other detailed comments are following:

Figures are small and too small font size is used. There is large use of panels and they are wrongly named (e.g. A and A1), with the name labels located inside the plot area (while it should stay outside up left at each panel). Authors could consider to unified panels together to improve clarity (for example in Figure 1 the two bar plots of each measurement could be grouped in the same panel, and different bar colour could be used to discriminate among N conditions). Less relevant information could be moved in supplementary figures (e.g. the quantile-quantile plots in figure 1).

In Figure 1, the x axes for plants under normal and low nitrogen are not the same. I guess this was done to optimize the plot sizes. However, to highlight a change or shift in the distribution of the measures taken, I think it will be clearer if the x axes will be the same for the two N regimes in each measurement.

Figure 4 is very difficult to read, the names of markers is maybe unnecessary at this magnitude and they could be indicated as line (not showing the name) or the chromosomes could be displayed with a narrow shape to leave more space to labels. At the contrary, the genetic distance could be indicated. It is also missing in the legend the information about the LOD score plot.

Line210. Did author test normal distribution of data presented in Figure1? If yes can they describe the test applied and the pvalue?

Line 398. It is unclear why candidate genes have been investigated only for qTNL2-1 and qTNN4-1. Also later in discussion

Other minor issues:

Please introduce the definition of “P1-type” and “P2-type” in the text at first occurrence and not later in the text.

It is unclear which (if any) of the previously identified QTLs associated to branching (lines 505-515) are overlapping the QTLs detected by authors in this work (chapter 4.2 in discussion).

Line 211. “Figure 1A,B1” is maybe “Figure 1A,B”

Reviewer 2 Report

     Author reports novel QTLs associated with shoot branching, and results may be valuable for rice researchers. On the other hand, this article do only single field trial and reproducibility is uncertain. It does not present the detailed conclusion of genes associated with shoot branching and seems to be a progress report. I think this article may be suitable for another publications in agronomy or genetic analysis, but not for in Genes.

     I also have some questions and comments about the experimental design and the chapter of discussion as described below.

(1)The results of this article seem to be derived from one-time cultivation. Usually, filed trials should be performed multiple times for confirming reproducibility.

(2)Each amount of various elements other than nitrogen also differ between two experimental fields.  It is uncertain all phenomena observed in this research arise only from the nitrogen condition.

(3)In Table S1 and S2, correlation level between TN and other traits differ depending on each nitrogen condition, and this may reflect the effect of including gene of QTL locus. Author should discuss more about that point. In addition, perhaps there is a mistyping at the end of Table legend.

'root dry weight under normal nitrogen cultivation'-->'root dry weight under normal nitrogen cultivation (RDWN)'

(4)All reports of individual gene function associated with shoot branching should be cited.

There is no citation about Os02g44360 and Os02g44370, this means no one study those genes previously?

(5)There are various overlapping explanations between 3.Results and 4.Discussion, and therefore article is unnecessarily long.  Brief communication is desirable.